



# Assessing methods for representing soil heterogeneity through a flexible approach within the Joint UK Land Environment Simulator (JULES) at version 3.4.1

Heather S. Rumbold[1], Richard J.J. Gilham[1], and Martin J. Best[1]

[1]Met Office, Exeter, Devon, EX1 3PB, United Kingdom

**Correspondence:** Heather S. Rumbold (heather.rumbold@metoffice.gov.uk)

**Abstract.** The interactions between the land surface and the atmosphere can impact weather and climate through the exchanges of water, energy, carbon and momentum. The properties of the land surface are important when modelling these exchanges correctly especially with models being used at increasingly higher resolution. The Joint UK Land Environment Simulator (JULES) currently uses a tiled representation of land cover but can only model a single dominant soil type within a grid box. Hence, there is no representation of sub-grid scale soil heterogeneity. This paper introduces and evaluates a new flexible surface-soil tiling scheme in JULES. Several different soil tiling approaches are presented for a synthetic case study. The changes to model performance have been compared to the current single soil scheme and a high resolution 'Truth' scenario. Results have shown that the different soil tiling strategies do have an impact on the water and energy exchanges due to the way vegetation accesses the soil moisture. Tiling the soil according to the surface type, with the soil properties set to the dominant soil type under each surface is the best performing configuration. The results from this setup simulate water and energy fluxes that are the closest to the high resolution 'Truth' scenario but require much less information on the soil type than the high resolution soil configuration.

## 1   Introduction

Land surface models (LSMs) are an important aspect of numerical weather prediction and climate simulations. They can be applied in both standalone and coupled mode, and on grids covering a range of spatial and temporal scales from high resolution weather forecasts (less than 1 km for several days) through to Earth System modelling for climate prediction (over 100 km for up to 100 years). Historically LSMs were considered to be a bottom boundary condition, the primary role being to provide fluxes of momentum, heat, moisture and carbon back to the atmospheric models. However, in last couple of decades an understanding of the importance and impact of land surface processes on the overlying atmosphere has grown. A number of studies have shown that the spatial variability of land surface properties have a direct impact on the surface fluxes (Eltahir and Bras, 1996; Eltahir, 1998; Betts and Ball, 1998; Hohenegger et al., 2009) and therefore need to be accounted for. The ability of LSMs to represent these properties at increasingly high resolution is limited by the resolution of the land ancillary information available (for example soil and vegetation types) and the computational expense of running at high resolution. Many land surface models have therefore adopted methods to represent sub-grid heterogeneity without the need for running at





high resolution.

In general there are three main approaches currently used to represent sub-grid heterogeneity: mosaic, tile and aggregated. The mosaic approach uses a sub grid to represent every pixel of land for which data is available. Each pixel has an appropriate set of parameters describing the physical behaviour of the surface and soil (e.g., soil texture, vegetation fractions, albedo, leaf area index, roughness length etc.). The tiling approach has the same parameter set but groups these pixels into a smaller number

of distinct surface types each with representative parameters. The aggregation approach attempts to represent the average properties of the grid box surface as a single set of representative parameters. However, Heinemann and Kerschgens (2005) highlighted that the terminology of the different approaches used (tile, aggregation, mosaic) is ambiguous in the literature and this is compounded by the need for different formulations imposed by different modelling architectures. This results in no one best approach being clearly recommended by the literature.

The community land surface model JULES (Joint UK Land Environment Simulator) is the land component of the Met Office Unified Model (UM) (Walters et al., 2019). JULES calculates the exchanges of energy and momentum between the surface and the atmosphere by representing a range of surface and sub-surface processes including snow, surface and soil hydrology, and vegetation physiology and dynamics. JULES currently uses a tiled model to represent surface heterogeneity with separate energy and water fluxes computed for each surface type within an atmospheric grid box (Essery et al., 2003). However, each of

the surfaces in the tiled scheme currently experiences the same sub-surface soil conditions, i.e. there is a single soil column per grid box. Due to the non-linear nature of soil processes, the dominant soil type is used for each grid box and soil parameters associated with this soil type are then used. The consequence of using this current method is that some of the spatial soil heterogeneity is lost when re-gridding from the ancillary source grid to the model grid and through selecting the dominant soil on the model grid. Therefore, a soil tiling approach that can represent the sub-grid scale soil heterogeneity should be beneficial.

Table 1 gives examples of LSMs used by the land modelling community and the approaches used by them to model sub grid soil heterogeneity. H-TESSEL uses a dominant soil texture class (i.e., course, medium, medium-fine, fine, very fine, organic) for every grid box, whilst CLM and ISBA have a single soil with properties aggregated within each grid box (Noilhan and Mahfouf, 1999). Most other LSMs have adopted more sub-grid scale soil tiling approaches. For example, the CLASS model has the capability of running with either a single soil column or one soil for each surface type, but applying grid box wide soil

properties in these cases (Li and Arora, 2012; Melton and Arora, 2014). Similarly, the NOAH model also assigns a soil column to each surface cover type but imposes identical soil properties for all tiles. The LM3, CABLE and the ORCHIDEE LSMs all assign a soil column to each surface cover type and allow different properties to be assigned to each.

With this in mind, this paper will describe and evaluate a new flexible surface-soil tiling scheme in JULES, which will allow sub-grid scale soil heterogeneity to be better represented. The general functionality of the scheme will be described, and a

synthetic case study will be used to define and test a range of possible new soil tiling approaches. The changes in model performance will be compared to standard JULES and a high resolution surrogate 'Truth' soil. The aim of this paper is to demonstrate the benefits of soil tiling in JULES by representing the sub-grid scale soil and surface heterogeneity in the most computationally efficient way.




## 2 Methodology

### 2.1 Scheme Description

The modular structure and component coupling within JULES has enabled a completely flexible surface-soil interface to be developed. The number of surface tiles in JULES depends on the land configuration being used (see (Walters et al., 2019)). However, most physical model configurations have nine surface tiles in each grid box, five of which are Plant Functional Types (PFTs) (broadleaf trees, needleleaf trees, C3 (temperate) grass, C4 (tropical) grass and shrubs) and four are non-vegetation types (urban, inland water, bare soil and land-ice). Each JULES surface tile calculates its own fluxes of heat, moisture and momentum, derived from bulk aerodynamic formulae through functions of specific humidity, air temperature, wind speed and available energy (see Sect. 2.1 in Best et al. (2011) for more detailed equations). These are averaged and weighted by the fractional cover of each surface tile over the grid box to produce grid box mean components of the surface energy balance.

The flux of water extracted by the vegetation from the soil for transpiration is determined by the root density and the soil moisture availability factor ($\beta$). $\beta$ is a multiplicative factor in the stomatal conductance equations that are used to calculate photosynthesis (equation 11, Best et al. (2011)). The root density is assumed to follow an exponential distribution with depth, with the depth scale varying between the different PFTs. $\beta$ is a dimensionless moisture stress factor, which is related to the mean soil moisture concentration in the root zone through the critical and wilting point factors (equation 12, Best et al. (2011)) and varies between zero and one. For soil moisture between saturation and the critical point, no limitation on soil evaporation or plant transpiration is applied (i.e., $\beta$ equals one). Below the critical point $\beta$ decreases linearly from one to zero at the wilting point, whilst below the wilting point no transpiration is possible (i.e., $\beta = 0$). The soil parameters used here (including the saturated soil water content, critical and wilting points) are calculated from linear equations relating soil moisture to the soil type. The soil types are based on spatially continuous textural properties (sand, silt and clay fractions) and the corresponding soil parameters are calculated using the pedeotransfer functions defined by Cosby et al. (1984).

Apart from those classified as land-ice, a land grid box can be made up of any mixture of surface types. A restriction under the current scheme is that there has to be either 100% coverage of land ice in a grid box or none, because land ice does not have its own prognostic water store. It uses the soil temperature profile to represent the thermal structure of the ice and moisture transport is neglected. As all surface tiles currently share the same soil information for temperature and moisture, this means it is not possible to have a fractional cover of land ice. A new soil tiling scheme could allow a fractional cover of land ice, giving this tile its own soil column and therefore enabling it to represent its own temperature and moisture profile separately from other surface tiles.

The urban surface tile is characterised by a large thermal inertia (one tiled scheme by Best (2005)) and is only in radiative exchange with the underlying soil. Therefore, the capacity of the urban tile to hold water is minimal and drainage of water is preferred over infiltration. This limits the evaporation to periods directly after precipitation and so the urban tile is therefore equivalent to a dry, one-layer block of concrete with a high heat capacity.

With the new scheme, each grid box has the capability to have a different number of surface tiles and soil tiles and the key feature is managing the connectivity between them. In JULES the surface is implicitly coupled to the atmosphere (Best et al.,





2004) and therefore needs to remain fixed at the resolution of the atmospheric driving data. This allows it to capture the
fast timescales of the turbulent processes and can sustain longer time steps for computational efficiency. The soil however,
responds to slower diffusive processes and hence can be explicitly coupled to the surface without encountering numerical
issues. This removes the limitation of the soil needing to be on the same grid as the surface and therefore can be modelled
at a higher resolution. As a result of this coupling, each surface tile operates in isolation, interacting with the atmosphere
through its own fluxes. Each soil tile also operates in isolation, interacting with the surface tiles above it through the exchange
of energy and moisture. There needs to be a precise mapping between the surface and soil tiles to enable them to exchange
information between them. This exchange is simple in cases where there is one-to-one mapping between the surface and soil
tile. Evapotranspiration is calculated using the soil moisture stress factor ($\beta$) from the soil tile under each surface tile. However,
in a grid box where there is a many-to-many interaction between surface and soil tiles, i.e. each surface can access more than
one soil column and visa versa, the weighted averages of the $\beta$ from the soil tiles in each grid box are used by each surface tile
to calculate evapotranspiration.

In this work, a synthetic case study has been used to define the surface-soil tile mapping which allowed the representation of
a wide range of different surface and soil tile arrangements (shown in Sect. 2.1.1). Standard JULES currently has nine surface
tiles and a single soil column. If each surface tile was then given its own independent soil tile (i.e. by tiling the soil according
the surface heterogeneity), then there would be nine surface tiles and nine soil tiles. It is also possible for each surface tile to
access multiple soil tiles or to share soil tiles with other surface tiles. In this case the interaction of the surface and subsurface
processes can become more complex, but the scheme provides the potential for a computationally optimal configuration with
all surface and soil tiles represented.

### 2.1.1 Experimental Set Up

In order to demonstrate the benefits of soil tiling, a range of different surface and soil tile arrangements have been tested using
a synthetic case study and meteorological forcing for a temperate mid latitude site. A single grid box has been generated using
an artificial mixture of surface and soil tile combinations (shown by Fig. 1), thus allowing many different tiling approaches to
be tested. The artificial grid box consists of 10 by 10 pixels, each with one of five different surface types and one of three soil
types, the intention being to capture the full range of surface and soil properties. The surface types correspond to commonly
used JULES surface types namely, broadleaf tree (BLT), needleleaf tree (NLT), C3 grass (C3G), urban (Ur), and bare soil (BS)
(Best et al., 2011). The three soil types are clay, loam and sand as defined by Cosby et al. (1984). The urban tile is included
here because the processes involved here make for a useful and interesting test for the new soil tiling scheme.

In order to apply the mosaic approach to the example in Fig. 1, it would require 100 surface tiles representing each pixel of
land cover with a one-to-one mapping to 100 soil tiles with the corresponding soil properties. This can be thought of as a higher
resolution grid for the surface processes compared to the atmospheric forcing, with each surface tile having its own separate
soil column. For this study, tiling approaches are being explored which will effectively group the properties of each pixel into a
125 smaller number of discrete categories. This approach is computationally more efficient than the mosaic approach and therefore
more appropriate for the modelling systems considered in this work. It is assumed at this stage that there isn't any interaction





between the different soil columns.

The five surface-soil configurations explored in this work are shown in Fig. 2. Figure 2a is the 'Default Configuration' (DC) currently used by JULES. Here a single grid box with a dominant loam soil (B) is shared between all surface tiles. There is a

one-to-one mapping between the surface and soil, so each surface type can effectively access all the moisture in that soil. If rainfall infiltrates into the soil via a particular surface tile, there is no constraint for it to be removed by the same surface tile.

Figures 2b and 2c allow each surface tile to have their own soil tile (i.e., tiling the soil according to the surface type) and this results in five surface types and five soil types. This allows a one-to-one mapping between surface and soil, so each surface type can effectively access water from one soil column only. In Fig. 2b (SurfGB), the soil is tiled by surface type, but each

soil tile has the same grid box dominant soil type (B, loam), and therefore have the same properties (as used by DC). This is an appropriate approach if the resolution of the soil data is less than that of the land cover data. In Fig. 2c (SurfDom), the soil is tiled by surface type, but each soil has the properties of the dominant soil type for that surface. Therefore, the soil tiles can be different and are not constrained to the dominant type for the grid box. This approach requires higher resolution soil information than the previous approach and allows a greater degree of soil heterogeneity within the grid box.

Figure 2d (HResTex) uses the higher resolution soil information to map all the possible combinations of surface and soil tiles. There are nine combinations of surface and soil in the artificial grid box, hence this requires nine surfaces and nine soils. This is a compressed version of a mosaic approach where the same surface/soil pairs are only calculated once.

Finally, Fig. 2e (HResTexAgg) represents the most computationally efficient method of representing all surface and soil types. Here each surface can interact with all of the soil types as required, and visa versa. Therefore, there are only the five surface

tiles and three soil tiles. The key difference for this approach is that there is a many-to-many interaction between surface and soil tiles, i.e. each surface can access more than one soil column and the fluxes can be distributed in a more complex way. For example, moisture infiltrating from the BLT surface tile will be distributed between the clay (A) and the loam (B) soil tiles. Similarly, soil moisture from the loam soil (B) can be used to supply evapotranspiration for both the BLT and NLT surface tiles.

Each of the five surface-soil configurations have been used to run JULES offline driven using forcing data from WFDEI (WATCH Forcing Data methodology applied to ERA-Interim data, Weedon et al. 2014). The WFDEI data spans the period of 1979 to 2012 and is available for all land points at 0.5 by 0.5 degrees resolution globally. Here JULES has been run for an inland location in England (52.25N, 0.25E), from 1980 to 2010, using the GL4 science setup (Walters et al., 2014). Note that this single UK site is not intended to be representative of all climatic regimes. It is used as a demonstration site and results

may therefore vary at other locations. Each tiling configuration was allowed to spin up until convergence of soil moisture for the first year of data (i.e. run for multiple years for the first year of data) and then run freely for the whole 30 year period. The output from these runs have then been compared and evaluated against each other based on their complexity and their ability to represent the 'true' heterogeneity of the grid box. Given this is synthetic grid box no observations are available. Therefore, the high resolution soil run (HResTex) is the closest thing we have to the truth and will be used to evaluate.





## 3  Evaluation of energy and moisture fluxes

In this section, the energy and moisture fluxes from each of the five surface-soil configurations are evaluated against output from the high resolution soil run (HResTex). This run uses the maximum possible amount of soil information available without using soil tiles and will therefore be used as a surrogate 'truth'. It is expected that the different soil tiling strategies will result in changes to the surface energy balance and soil moisture due to the way in which energy and water are partitioned between the soils and surfaces. Results will also be compared back to the current single soil scheme.

The impact of the different soil tiling methods on heat and moisture fluxes is shown in Fig. 3. Plotted here are the 30 year monthly mean surface temperatures, latent and sensible heat fluxes and total change in soil moisture, averaged over the grid box for each method. The lines for SurfDom and SurfGB are not shown because they overlap with the HResTex run in all variables. The vertical bars represent one standard deviation from the DC (solid) line indicating the range in annual variability. All runs remain within one standard deviation indicating that the annual averages for all experiments are within the annual variability of the default configuration run.

The impact on average mean surface temperature is much smaller than the basic measure of inter-annual variability used here. The importance of variations in this quantity is acknowledged but we note that different users of LSMs will be concerned about different magnitudes and timescales of variation. Therefore, we do not consider surface temperature further. The impact on latent and sensible heat fluxes is much larger especially from May to August. Similarly the soil moisture change also shows large variations between the methods from April to August, as well as additional smaller variations from October through to January. The sensitivity of the fluxes to variations in soil moisture is greatest when the soil is unsaturated. Therefore, from April to August the fluxes are most likely to be impacted by the increased soil moisture limitations on evapotranspiration which allows more variability. The peak in latent heat flux for the DC (solid line) run is up to $10 \, \mathrm{Wm^{-2}}$ greater than the high resolution run (HResTex, thick dashed line) during June and July (the opposite is the case for the sensible heat flux). The SurfGB and SurfDom experiments (not shown) have a similar annual cycle to HResTex but use far less soil information and tile the soil according to the surface type (as opposed to using high resolution soils). The more complex, HResTexAgg experiment (dotted dashed line), shows a smaller increase in the peak latent heat flux and is much closer in magnitude to the DC experiment, especially in Autumn. The change in total soil moisture across all model runs shows a notable dry down in soil moisture from April to August indicated by the negative change in total soil moisture (bottom right plot). For the DC run the dry down in soil moisture still continues throughout April, May and June. During this time HResTex, SurfGB, SurfDom and HResTexAgg runs have a much slower rate of dry down and the decrease in soil moisture stays constant. From July onwards the dry down rate is decreasing across all runs until September when soil moisture increases again in all cases. DC and HResTexAgg then proceed to moisten faster from October onwards. Given that these runs both lost more soil moisture in the summer, they are still slightly drier than HResTex but are gradually become less dry over the winter.

Figure 4 shows the mean annual cycle for the soil moisture stress factor ($\beta$). Each row in the figure represents the soil layers one to four and each column represents the experiments with decreasing soil heterogeneity from left to right. Each line represents a soil tile with colours representing the surface tiles (BLT Green, NLT Blue, C3 red, Ur purple and BS black) and line styles





representing the soil tile type (Clay dotted, loam solid and sand dashed). For HResTexAgg, there are multiple surface types for
each soil type (as per Fig. 2e) hence lines are black and soil tiles are represented by the above line styles. As there is only one
soil for DC, there is just one line here for each soil layer. The bottom row shows the grid box mean and layer average $\beta$ for
each experiment.

In general soil tiling is allowing a greater degree of variation in the soil moisture stress through adding more soil tiles and
through a dependence of $\beta$ on the soil type. The differences between DC and SurfGB are due to soil tiling but not due to soil
type (soils are the same). For the DC experiment, the $\beta$ values for all four soil layers are below one and decrease to 0.4 during
the summer (0.5 for layer four). When compared to the profiles from SurfGB, SurfDom and HResTex, $\beta$ can vary much more
for the different soil tiles. The annual range in $\beta$ is from 0.3 to 1.0 in layer one and 0.2 to 1.0 in layer four. This is because
the addition of soil tiles (and therefore more soil columns) has allowed each surface tile to have different rooting profiles and
rates of water extraction. The resulting grid box mean $beta$ for SurfGB is lower than DC, which can explain the reduction in
latent heat flux shown by Fig. 3. The small differences between SurfDom and SurfGB are due to differences in the soil type
only as the number of soil tiles is the same in both runs. SurfDom has BT and C3 grass with clay soils compared to the loam
soils from SurfGB. These soils lead to a slightly lower $\beta$ over all levels and lead to a small reduction in the grid box mean $\beta$.
HResTex shows that $\beta$ is very dependent on the soil type. However, there is a lot of variation occurring between the different
levels and between the different surface-soil tile combinations. The differences are very much connected to where the plant
roots (determined through the surface tile) are and how available the soil water is for that type of soil. Clay soils typically hold
more water, compared to sandy soils, but this water is held tightly within the pore spaces, leading to lower drainage and higher
wilting and critical points, hence soils become water stressed more easily. In contrast, the loam tiles have considerably higher
$\beta$ and remain closer to the critical point. Despite this huge amount of variation between the surface-soil tile combinations,
the grid box mean $\beta$ shows a similar pattern to SurfDom and SurfGB implying that adding extra soil information makes
little difference to soil moisture stress and therefore the evaporative fluxes and latent heat flux. Unlike the other experiments,
HResTexAgg has a complex overlap between surface and soil tiles and exhibits very different $\beta$ characteristics. In particular,
the clay soil tile (black dotted line) has a much lower $\beta$ than the other tiles, especially in layer four. $\beta$ under the urban tile stays
at one suggesting that this combination of surface and soil tiles has no impact on the soil moisture.

Figure 5 shows the average annual cycle of grid box mean soil moisture in each layer for the SurfGB (dashed line) and DC
(solid line) experiments. The figure shows that SurfGB tends to have more soil moisture in the summer and less in winter
compared to DC. This becomes more emphasised in layer four with larger differences between the runs at the peak in April
and through into October. This result is consistent with the lower latent heat fluxes observed in Fig. 3. Less latent heat flux
implies less water extraction from the soil and therefore a wetter soil. The reason these differences in water extraction are
occurring is due to the mapping between soil and surface tiles.

The variations between the experiments seen in Fig.'s 3 to 5 are due to differences in the amount of water that the vegetation is
able to extract through the roots from soil moisture. This depends on whether the water in the soil column is shared between
surface tiles and how the different soil characteristics are represented through the soil tiling. The DC experiment has a single
soil with multiple surface tiles. These surface tiles are able to extract soil moisture from all soil layers at an even rate, with no





one soil layer drying out preferentially. Overall this means more soil moisture is lost in total, hence the onset of soil moisture

stress is slowed, and the dry down can continue for longer. This leads to a higher peak in latent heat flux shown in Fig. 3. In contrast, SurfGB, SurfDom and HResTex all have a one-to-one mapping between the surface and soil tiles. In each case the surface tile will only be able to access water from the soil column underneath through one rooting profile defined by the surface tile. This results in those tiles with greater root density drying quicker (than a single soil column), leading to an earlier onset of soil moisture stress compared to the DC experiment. The dry down is therefore shorter and the peak in latent heat flux is lower.

Soil tiling is also allowing a greater degree of variation in $\beta$ (shown by Fig. 4). This is partly because more variation in the soil type has been introduced by the addition of more soil tiles. However, in addition to this, surface tiles can now access their own soil columns and can therefore extract water from each layer at different rates. This process on its own introduces more variability in $\beta$ between soil columns of the same soil type but different surface type. The differences in $\beta$ between DC, SurfGB, SurfDom and HResTex are due to variations in the root densities and water extraction between the PFTs. If the surface

is a grass tile then the majority of the roots (greater than 90%) are in the top 1 metre, so water is preferentially extracted from layers one and two. This leads to a larger drop in $\beta$ during the summer in these layers. If the surface is a tree tile, then the majority of roots are much deeper (83% of BLTs and 69% of NLTs have roots in layers three and four) and water is mostly extracted from the deepest two layers of the soil profile. This results in $\beta$ staying constantly lower throughout the year in layer three and four for all three experiments. If the surface is bare soil, water can only evaporate from the top layer (i.e. the top

10cm), hence $\beta$ drops to below 0.5 in layer one over all the experiments. Where the $\beta$ drops below one in layers two to four this is more likely to be due to differences in soil type because there isn't any water extraction from roots. Differences in soil type will impact how the soil moisture will drain under gravity and how easily the soil will become stressed.

Differences in the soil-surface mapping have meant that the different surface tiles are now able to extract water at different rates independently and it is this which has led to the large variation in $\beta$ across the soil tiles. The preferential drying of layers with

the highest percentage of roots and largest fraction of surface type has meant that lower values of $\beta$ have been reached during the summer months compared to the DC experiment. This has started to feedback on the amount of water that is extracted for evapotranspiration, resulting in a slowing of the dry down and at the same time an increase in soil moisture (show by Fig. 5). Soil tiling is therefore acting in a way to regulate soil moisture loss and therefore evapotranspiration when $\beta$ is low.

HResTexAgg is a more complex case with many-to-many interactions between surface and soil tiles i.e. each surface can access

more than 1 soil and the soils can have different characteristics. This leads to less preferential drying of layers because there is a more even distribution of roots between the surface types, similar to the DC. The intricacies of the surface-soil mapping in this experiment have led to results that are an intermediate case between the DC and the other experiments. Figure 4 is showing very different $\beta$ characteristics to the other experiments. In particular, the clay soil tile (black dotted line) has a much lower $\beta$ than the other tiles. Compared to sandy soils, clay soils hold more water but this water is held tightly within the pore spaces,

leading to lower drainage and higher wilting and critical points, hence soils become water stressed more easily. In contrast, the loam (red dotted line) and sand (green dotted line) tiles have considerably higher $\beta$ and remain closer to the critical point. In this case because the surface types are interacting with more than one soil tile, the $\beta$ values are combined in order to supply the surface tiles with a single soil moisture stress factor. Taking the average $\beta$ results in a value which is higher than the clay





soil tile value but lower than the sand and loam soil tile values. Hence, too much water is extracted from the stressed clay tile
and too little from the less stressed sand and loam. This sets up a positive feedback which acts to maintain this difference.
Therefore, although the HResTexAgg method does add value over the DC, it doesn't do as well as SurfDom and SurfGB at
reproducing the high resolution soils experiment, HResTex. It also demonstrates that the non-linear interactions need to be
managed correctly in order to gain the benefits of additional soil heterogeneity.

## 4  Conclusions

This paper describes and evaluates a new flexible surface-soil tiling scheme for the land surface model JULES, which allows
sub-grid scale soil heterogeneity to be better represented. The functionality of the scheme has been described and is used to
assess potential methods for improving soil heterogeneity.

A synthetic case study has been used to define and test a range of possible new soil tiling methods. Three soil tiling methods
have been considered here. Two of these tile the soil by surface type where each surface has its own soil tile (using either
the grid box dominant, SurfGB or the surface tile dominant soil, SurfDom). A third method is a many-to-many approach
(HResTexAgg) whereby the surface can interact with all of the soil types as required, and visa versa. The changes in model
performance have been compared to the single shared soil scheme currently used by JULES (DC) and a high resolution sur-
rogate 'truth' soil (HResTex), which uses higher resolution soil property data to map all possible combinations of surface and
soil tiles.

Overall, soil tiling introduces a decrease in monthly mean latent heat flux (increase in sensible heat flux) compared to the
standard DC run, with the largest differences observed from May to August. Soil tiling methods that tile the soil according
to the surface type (SurfGB and SurfDom) have almost identical fluxes to HResTex. The annual cycle of the change in total
soil moisture shows a notable dry down from April to August. The SurfGB, SurfDom and HResTex runs are comparable and
show a much slower rate of dry down compare to the DC run. These runs also show a much greater degree of variation in soil
moisture stress compared to the DC run.

The variations between the experiments are due to differences in the amount of water that the vegetation is able to extract
through the roots from soil moisture and this is dependent on the type of soil and the distribution of roots in the soil (which
depend on the surface type). The DC experiment has a single soil with multiple surface tiles. These surface tiles are able to
extract soil moisture from all soil layers at a more even rate, with no one soil layer drying out preferentially. Hence, the onset
of soil moisture stress is slowed, and the dry down can continue for longer, leading to a higher peak in latent heat flux and
more soil moisture lost in total. In contrast, SurfGB, SurfDom and HResTex all have a one-to-one mapping between the surface
and soil tiles. The surface type will only be able to access water from the soil column underneath through one rooting profile
defined by the surface type. This results in certain layers with greater root density drying quicker, leading to an earlier onset of
soil moisture stress compared to the DC experiment. The dry down is therefore shorter and the peak in latent heat flux is lower.
These differences in water extraction rates have led to the large variation in $\beta$ across the soil tiles. The preferential drying of
layers with the highest percentage of roots and largest fraction of surface type has meant that lower values of $\beta$ have been





reached during the summer months compared to the DC experiment. This feeds back on the amount of water that is extracted for evapotranspiration, resulting in a slowing of the dry down and at the same time an increase in soil moisture. Soil tiling is therefore acting to regulate grid box mean soil moisture loss and evapotranspiration when $\beta$ is low.

The more complicated, aggregated approach (HResTexAgg) requires more soil information and has complex interactions between the soil and surface tiles, but the reduction in latent heat flux is less. For this case, the extraction of water from the soil tiles does not yield the correct transpiration due to the non-linear combinations of the soil moisture stress for vegetation from each soil type. This results in too much water being extracted from some soils and too little from others.

Despite the increase in heterogeneity between SurfGB, SurfDom and HResTex, the results from these experiments are very

similar suggesting that it is the changes to the way vegetation accesses the soil moisture that is more important, rather than the variability added by the soil heterogeneity itself (although this is still a factor). This demonstrates that any of these three methods would be appropriate to represent sub-grid scale soil heterogeneity in JULES. However, tiling according to the surface type and using the dominant soil for that surface type (SurfDom), gives the best compromise between giving the biggest positive impact without requiring very high resolution soil information (like HResTex). It still allows some heterogeneity between

soil tiles (unlike SurfGB), which is important for other hydrological soil processes (such as lateral flow) and is the closest to the high resolution 'Truth'. Overall this method provides the most flexibility and is the most computationally efficient way to introduce sub-grid scale soil heterogeneity into JULES.

There are many applications which will be improved by the addition of this scheme. For example, Smith et al. (2022) demonstrates the importance of soil tiling for simulating discontinuous permafrost and uses a simplified form of the code to address

biases in methane emissions. This study also uses lateral soil moisture flow to improve the snow depth, soil moisture and temperature over the permafrost landscape. Another application is the recent implementation of the lake model, FLake (Rooney and Jones, 2010) into standalone JULES. It has had issues with correctly diagnosing the soil temperatures underneath the lakes. Soil tiling by surface type would allow lake tile soils to be thermally coupled in a different way to standard soil-atmosphere coupling, such that soil temperatures could be adjusted to deal with depth. The new scheme will also provide the opportunity

to introduce a fractional cover of land ice. Under the existing set up there has to be either 100% coverage of land ice in a grid box or none, because land ice does not have its own prognostic water store. The new scheme could give the land ice tile its own soil column and therefore enable it to represent its own temperature and moisture profile separately from other surface tiles.

The work in this paper has not yet considered the impact of lateral flow of soil moisture between soil columns which will become more significant as resolution is increased. By allowing lateral flow, the soil profiles within a grid-box could exchange

water through slope and diffusive processes, thus representing hydrological variability in a more realistic way. This could impact on the accuracy of the surface fluxes at sufficiently high resolution and may have wider significance when used within a coupled weather and climate model. Hence, it is possible that the inclusion of lateral flows could influence the conclusions of this study, and should be considered for future experiments.



*Code and data availability.* The JULES model code is freely available to anyone for non-commercial use from the Met Office Science
Repository Service (MOSRS) (https://code.metoffice.gov.uk/ last access: 3 August 2022). Registration is required and is subject to comple-
tion of a software licence (for details of licensing, see https://jules.jchmr.org/content/code, last access: 3 August 2022). Visit the registration
page (http://jules-lsm.github.io/access_req/JULES_access.html, last access: 3 August 2022) to request code access and a MOSRS account.
The results presented in this paper were obtained by running JULES from the following branch: https://code.metoffice.gov.uk/trac/jules/brow
ser/main/branches/dev/heatherashton/vn3.4.1_soil_tiling?rev=23611. This is a development branch of JULES version 3.4.1 which includes
the new surface-soil tiling scheme. This branch can be accessed and downloaded from the Met Office Science Repository Service once
the user has registered for an account, as outlined above. The input and output data and plotting scripts used in this paper are provided at
https://doi.org/10.5281/zenodo.6954142 (Rumbold et al., 2022). The experiments described in this paper uses the configurations prescribed in
the branch here: https://code.metoffice.gov.uk/trac/jules/browser/main/branches/dev/heatherashton/vn3.4.1_soil_tiling/configurations?rev=23
611. The "Gzipped" WFDEI files are freely available from the WATCH ftp site at IIASA, Vienna (online: ftp://rfdata:forceDATA@ftp.iiasa.ac.at
and click on /WATCH_Forcing_Data and then /WFDEI, or for ftp downloads: site = ftp.iiasa.ac.at, username = rfdata and password = force-
DATA, then use: cwd/WFDEI). The /WFDEI directory includes files listing grid box elevations and locations. An alternative source of
WFDEI data is provided by the DATAGURU website for climate-related data at Lund University (http://dataguru.nateko.lu.se/, log in, then
go to 'Application'). More information is available from Weedon et al. (2014) in Sect. 6, pages 8 and 9. The script available at Rumbold et al.
(2022) extracts the WFDEI data for the single latitude/longitude point used in this paper.

*Author contributions.* MJB initiated the research into implementing a sub grid scale soil heterogeneity scheme within JULES and provided
general scientific guidance throughout the paper. RJJG and HSR scoped out and wrote the code, while RJJG set up and ran the experiments.
HSR did the analysis and wrote the paper.

*Competing interests.* The contact author has declared that neither they nor their co-authors have any competing interests.

*Acknowledgements.* This work was supported by the Joint BEIS and Defra Met Office Hadley Centre Climate Programme (GA01101). The
authors thank the reviewers and editor for their comments and suggestions. The authors also thank Adrian Lock for helpful comments on
preparing this manuscript.





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





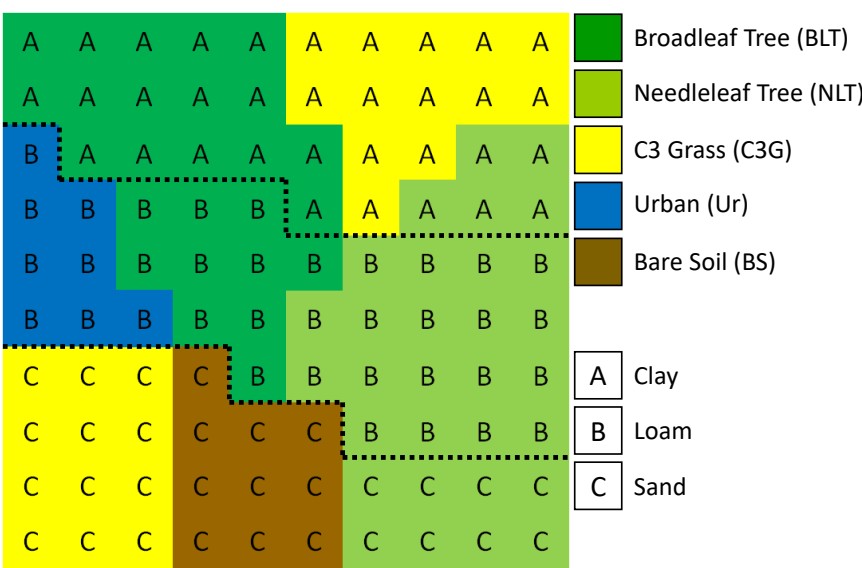

**Figure 1.** Synthetic grid box used to define and test the new soil tiling configurations. This consists of a 10 by 10 pixel grid, each with one of five different surface types and one of three soil types. The surface types correspond to broadleaf tree (BLT, dark green), needleleaf tree (NLT, light green), C3 grass (C3G, yellow), urban (Ur, blue), and bare soil (BS, brown). The three soil types are clay (A), loam (B) and sand (C) as defined by Cosby et al. (1984)





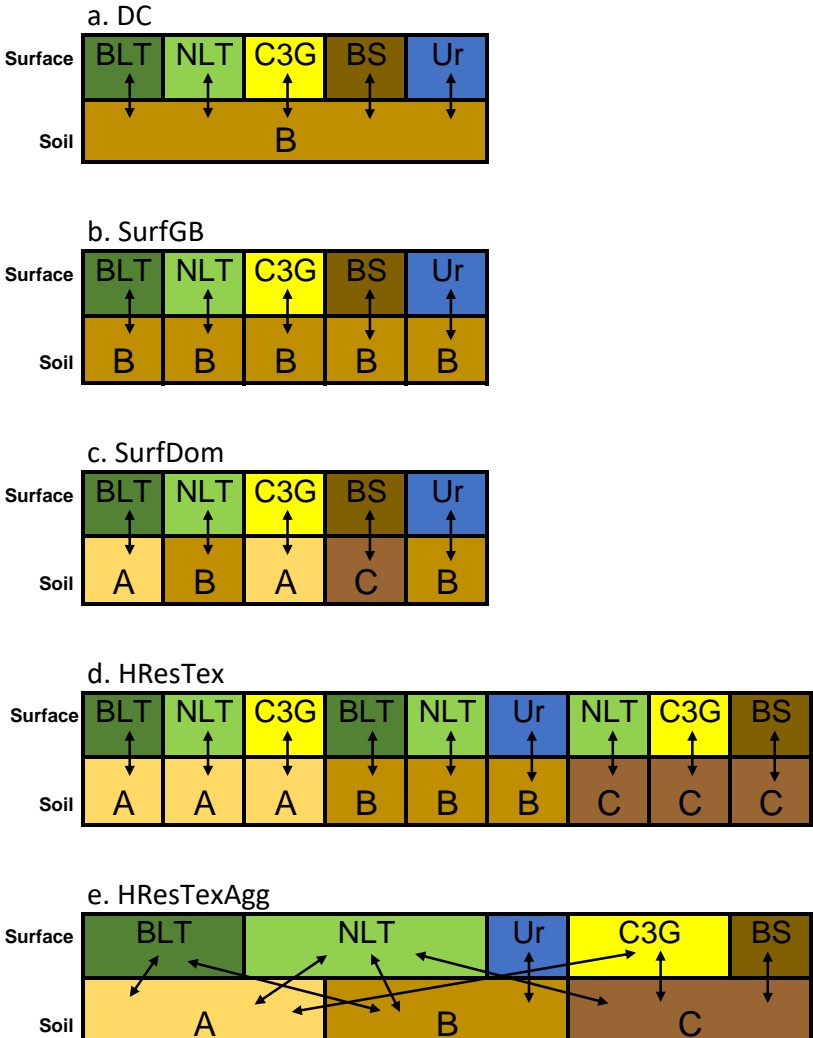

**Figure 2.** Schematic showing the 5 surface-soil configurations used in this work (abbreviations as per figure 1): (a) The 'Default Configuration' (DC) currently used by JULES. (b) 'SurfGB', shows the soil is tiled by surface type with each soil tile having the same grid box dominant soil type. (c) 'SurfDom', shows the soil is tiled by surface type with each soil having the properties of the dominant soil type for that surface. (d) 'HResTex' uses the higher resolution soil information to map all the possible combinations of surface and soil tiles. This configuration is considered to be the surrogate truth in the absence of observations. (e) 'HResTexAgg' is the fully compressed version of the mapping and shows each surface type can interact with multiple soil types and visa versa.

**Figure 3.** 30 year monthly means averaged over the grid box for surface temperature, latent and sensible heat fluxes, and the change in total soil moisture. Thick dashed line: HResTex, dashed dotted line: HResTexAgg and solid line: DC. Bars are one standard deviation from DC line.

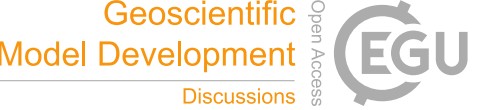



**Figure 4.** Mean annual cycles for soil moisture availability factor. Each row corresponds to a soil model level. Colours represent surface tiles and line styles represent the soil tile type. For HResTexAgg, there are multiple surface types for each soil type (as per figure 2e) hence lines are black and soil tiles are represented by the above line styles. Final row shows the grid box mean and layered average soil moisture availability factor





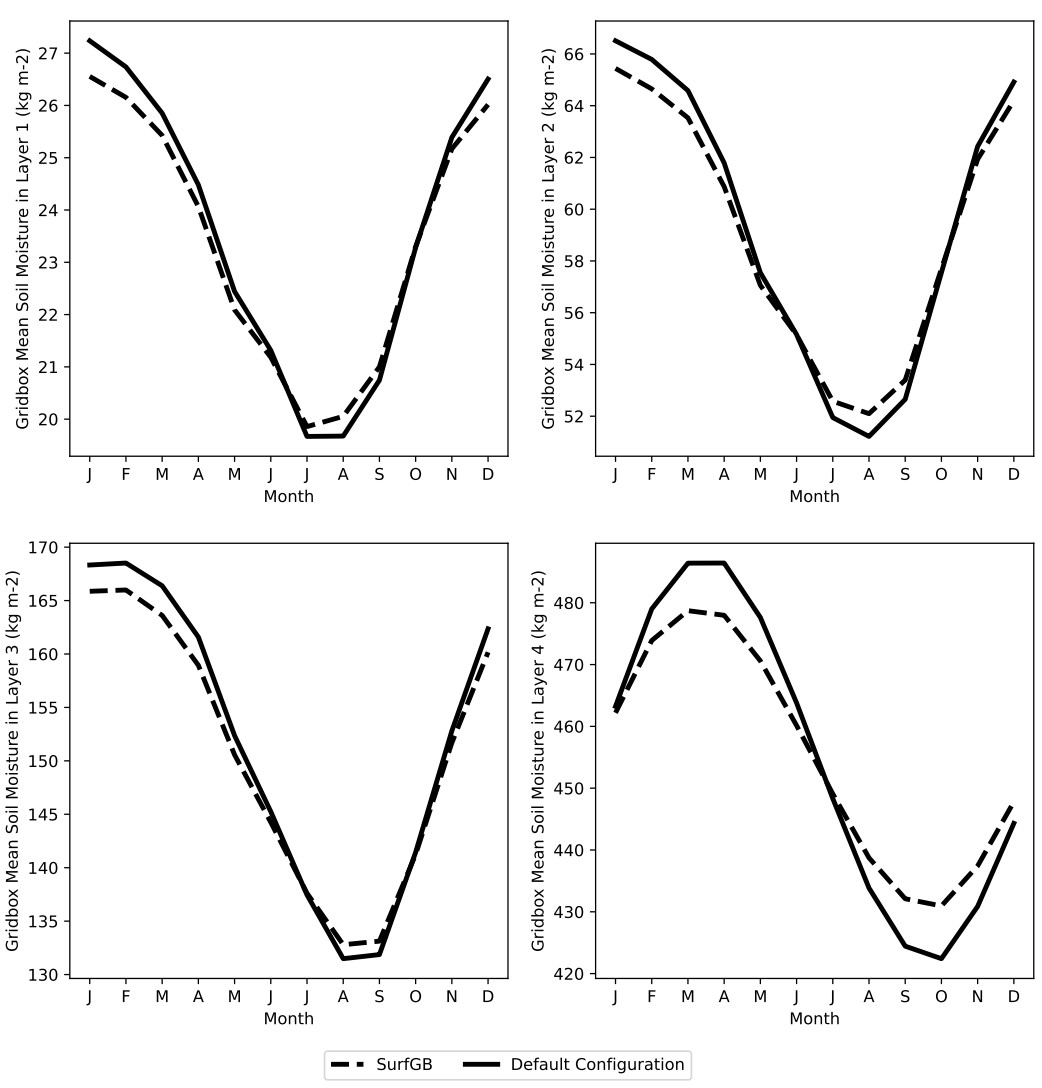

**Figure 5.** Grid box mean annual cycles of soil moisture in model layers for SurfGB (dashed line) and DC (solid line).





**Table 1.** Examples of currently used Land Surface Models with their methods for representing sub grid scale soil heterogeneity

| Model | Institution | Reference | Soil Tiling Method |
|---|---|---|---|
| NOAH integrated land model | GFDL | Ek et al. (2003) | Soil per surface type, identical soils |
| H-TESSEL | ECMWF | Balsamo et al. (2009) | Single dominant soil texture class |
| CLASS | ECCC | Verseghy (1991, 2000) | Single soil with properties aggregated or soil per surface type |
| ISBA | Meteo-France | Decharme and Douville (2006) | Single soil with properties aggregated |
| LM3 | U.S. Geological Survey | Milly et al. (2014) | Soil per surface type, different soils |
| ORCHIDEE | IPSL | Ducoudré et al. (1993); de Rosnay (2003) | Soil per surface type, different soils |
| CABLE | CSIRO/BOM | Kowalczyk et al. (2013) | Soil per surface type, different soils |
| CLM | NCAR | Oleson et al. (2013) | Single dominant soil |
| JULES | Met Office | Best et al. (2011); Clark et al. (2011) | Single dominant soil |