# Peer review of "Assessing methods for representing soil heterogeneity through a flexible approach within the Joint UK Land Environment Simulator (JULES) at version 3.4.1"

_Geoscientific Model Development, 2022_

## Author Comment (AC1)

**Reply to the reviewers' comments: gmd-2022-139**

Heather S. Rumbold et al.

November 1, 2022

We would like to thank the reviewers for their time to read and comment on this manuscript. We have addressed all the revisions, and hope this improves the manuscript satisfactorily. Blue text below is our response to the reviewers' comments (reproduced in black).

**Reviewer Comments 1**

**General comments**

The paper looks at how sub-grid scale soil heterogeneity can be added to a complex land-surface model. Using a synthetic example, different configurations are tested, exploring both increased heterogeneity and computational efficiency.
It is an interesting development shown to have an important impact on model vegetation-soil moisture interactions, especially at high resolutions. The paper is well written and a good fit for GMD. My comments are mainly about clarifying parts of the manuscript.

**Specific comments**

How was the synthetic example created? As in, how were the different fractions of PFTs and soil textures chosen? Is it based on the UK site where the meteorological forcing was chosen?
The synthetic example grid box is artificial and devised to enable us to represent all five different surface types and a full range of soil textures. The fractions were chosen to give even spread of the different soil-vegetation combinations. If we had based this on a real life case study, over the similar sized grid box, then it is unlikely that we would have observed such a range in vegetation and soil properties. This would have limited us from fully testing the capacity of the parameterisation.
We have added the following line to the manuscript to clarify this point (lines 121 - 126) - "A single grid box has been generated using an artificial mixture of surface and soil tile combinations (shown by Fig. 1). This was devised in order to represent all five different surface types and a full range of soil textures. The fractions were chosen to give even spread of the different soil-vegetation combinations, allowing a range of different tiling approaches to be simulated and therefore fully testing the capacity of the parameterisation. The artificial grid box consists of 10 by 10 pixels, each with one of five different surface types and one of three soil types."

What resolution is the full grid box meant to represent? 0.5 degrees like the forcing?

That is correct, the resolution of the grid box is 0.5 degrees with a sub grid of 10 by 10 pixels. We have now made this clear in the manuscript.

For HResTexAgg, how are the interactions distributed? For example, it is mentioned that moisture infiltrating from BLT is distributed between the clay and loam. Is this distribution even, i.e., 50:50? Or is it proportional to the fraction of soil texture? i.e., 16/26 to clay and 10/26 to loam in this case.

For HResTexAgg the moisture is distributed from the surface to the soil as a proportion of the soil texture i.e., for BLT, 16/26 goes to clay and 10/26 to loam and similarly for other surface tiles. We have clarified this detail further in the manuscript.

L180: Can the authors comment more on the fact that SurfGB and SurfDom match HRexTex in Fig. 3? I realise this is discussed more at the end of this section, but I think a sentence here explaining how they all have a one-to-one mapping would help the reader.

We have added the following text to explain this further (lines 192 - 196): "The SurfGB and SurfDom experiments (not shown) have a similar annual cycle to HResTex despite the fact that they use far less soil information and tile the soil according to the surface type (as opposed to using high resolution soils). The reason for this similarity is that they all use a one-to-one mapping between the surface and soil tiles. This results in similar rates of drying, onset of soil moisture stress and resultant latent heat fluxes."

Throughout the plots and analysis, four layers are discussed. However, I don't think the concept of layers is introduced. How many layers total make up the soil column in JULES? How deep and thick is each respective layer?

We have now added the following text (lines 84 - 86) - "The default number of soil layers in JULES is four with thicknesses 0.1, 0.25, 0.65 and 2.0 m, giving a total soil depth of 3 m. This configuration is designed to correctly capture the variation of soil temperature from sub-daily to annual timescales (Best et al., 2005)".

Since it is a synthetic example, it cannot be evaluated against observations. However, maybe the authors could comment in the conclusion on how future work could use observations. Furthermore, only one climate is tested (mid-latitude temperate). Could the authors comment on how the results would change for a different climate? For example, what does one might expect results to be for an arid site?

We have added the following paragraphs to the conclusion to discuss the use of observations and the possible impact of different climates (lines 357 - 371 ): "Given a synthetic example case study is used in this work, it cannot be evaluated against observations. In order for observations to be used, suitable resolution observation data is required. However, area-representative observations of soil moisture and surface fluxes (latent and sensible heat flux) are challenging to measure at the grid scales of interest here. Many measurements of soil moisture and surface fluxes are at a single point which may not be representative of the surrounding area. Soil moisture is spatially heterogeneous whilst the surface fluxes are dependent on the footprint area. On the other hand, observations of soil moisture using satellite microwave sensing can provide

integrated values of near-surface soil moisture at resolutions of hundreds to thousands of square kilometres, but they are unable to penetrate more than a few centimetres into the soil (and therefore do not represent root zone soil moisture) and the signal can be masked by snow or dense vegetation. Given this lack of suitable observations, a high resolution (atmosphere, surface and soil) simulation could be used as surrogate observations in future work. Additionally, only one climate is tested here (mid-latitude temperate). It is possible that the conclusions of this study could change under different climates. For example, the energy and water fluxes at an arid site are likely to be less sensitive to variations in soil moisture and therefore soil heterogeneity would be less important. The impact of using other climates should be considered for future work."

**Technical corrections**

Throughout: change quotes ' to '
Done
Throughout: sub grid vs sub-grid
Sub-grid is now being consistently used throughout the manuscript
L18: in the last couple
Done
L26: tiled
Done
L30: Do you mean representative parameter values? Or additional parameters on top of the parameter set used in the mosaic approach?
Yes, we mean representative parameter values. Updated in the manuscript.
L62: remove extra brackets around the citation
Done
L102, L107, L146, L156: add missing , after i.e. to be consistent with the rest of the text
Done
L126: is not
Done
L184: autumn is not a proper noun
Done
L194: Clay does not need to be capitalised
Done
L204: "\" missing in the latex maths mode for beta
Done
L225: Is "Fig.'s" the correct shortening for multiple figures?
The guideline state that "The abbreviation 'Fig.' should be used when it appears in running text and should be followed by a number unless it comes at the beginning of a sentence" - There is no mention of how to refer to multiple figures in these guidelines, so we have changed this incidence to "figures" as it reads clearer.
L266: does not
Done
L293: These results
We do not think this change is required, so have left it as it is. The sentence "This results in certain layers with greater root density drying quicker," reads as it should

Fig.s 3 and 5: superscript is needed for the units
Done - updated plots
Fig. 4: beta as a symbol?
Done - updated plot

**Reviewer Comments 2**

**General comments**

Rumbold et al. describes improvements to the soil tiling scheme at different levels of complexity in the land surface model JULES. The preferred soil tiling scheme is determined by balancing the available resolution of soil types/surface types in a grid box and computational time required. A synthetic example grid box located in the UK is used to illustrate the effect on energy and moisture fluxes using the four different improved soil tiling methods along with the original simplistic method. Conclusions about the required complexity of the soil tiling scheme is mode based on this synthetic example grid box.

The improvement to the model is a highly desirable one and the methods and experimental setup is described in a clear and logical manner. Some improvements to the text should be done, especially avoiding repetitions. To the reader not familiar with the detailed representation of vegetation in JULES, some added information in this regard would help to ascertain the theoretical maximum complexity of soil/surface connectivity.

**Specific comments**

The complexity of the vegetation in the model should, to some extent, determine the ideal level of complexity of the soil tiling. How is vegetation demography represented in JULES? Would the different pixels within a surface type in Fig.1 have different age distributions or different land use histories? If yes, the "mosaic" soil tiling approach would perhaps be the ideal alternative, not considering data availability and computational limits. If no, the tiling approach would be enough. It would be helpful for the general reader if some information regarding this question was added.

Vegetation demography is not currently represented in JULES, but there are plans for the Robust Ecosystem Demography (RED) model to be included in JULES in the near future. Despite this development, it is currently not possible for JULES to run with soil tiling and vegetation demography (or dynamic vegetation through TRIF-FID). This is due to the code complexities of managing the evolution of vegetation fractions over each soil type, as well as ensuring the correct soil-vegetation mapping is maintained over time. At the current time this is beyond scope of the work presented here but is something we would like to consider for future work.

Can the version used, 3.4.1, be related to newer versions, which seem to include e.g. managed forests (?) and cropland as well as land-cover/land-use change ? Does v.3.4.1 contain any of these capabilities ? As already mentioned, these features would seem to influence the necessary complexity of the surface/soil tilling scheme.

The results presented in this paper were obtained by running JULES from a development branch at version 3.4.1 which includes the new surface-soil tiling scheme. The

results from this paper were then consolidated and used to develop an improved scheme that was formally implemented into the JULES trunk at version 5.7. The capabilities you mention above are likely to be available at version 5.7 so in theory they could be used with the new surface-soil tiling scheme. However, due to the code complexities mentioned in the previous response this may not be possible yet and almost certainly will not have been tested.

What is the temporal resolution of the processes in the model? It is not described (or it evaded me) how the consumption of water by vegetation from a single soil tile works with multiple surface tiles as in the DC and HResTexAgg options. A naive reader's guess could be that the interactions with the different tiles are done sequentially in the code, which would make the question of temporal resolution relevant. (Noticing that this is a special JULES issue, I realize that some of these questions are probably answered by other articles in this issue, so they may not be so critical.)
The driving data has a temporal resolution of 3 hours. However, JULES was run with a shorter time step of 30 minutes (with appropriate interpolation between data time steps) in order for the numerics to remain stable. The extraction of water from a single soil by multiple surfaces occurs instantaneously within the same time step. Details of the soil moisture extraction calculations can be found in section 4.2 in the Best et al., (2011) paper.

Why are not carbon fluxes considered ? Is it because the focus is on meteorological rather than climate applications? This should be mentioned.
That is correct - The focus of the paper was to evaluate the new surface-soil tiling scheme in JULES within a physical land configuration (i.e. meteorological rather than climate). Testing within the earth system context was beyond the scope of this current paper due to the code complexities mentioned above and incompatibility of the scheme with the dynamic vegetation.

I'm not sure the format of this journal requires it, but ideally, example grid cells from other climates would seem necessary. Unless the intended usage would be for the UK only, but this should be mentioned in that case.
We have added the following paragraph to the conclusion (lines 367 - 371): - "Only one climate is tested here (mid-latitude temperate). It is possible that the conclusions of this study could change under different climates. For example, the energy and water fluxes at an arid site are likely to be less sensitive to variations in soil moisture and therefore soil heterogeneity would be less important. However, the context of this work focused on testing the schemes limits and capabilities using an artificial grid box and therefore the impact of using other climates will be considered for future work."

Also ideally, real-case example sites using real land cover and soil type data at different resolutions would be beneficial to illustrate the significance of selecting one soil tiling scheme over the other.
We agree that using real land cover and soil type data would be useful to evaluate the soil tiling methods further. Work is currently in progress exploring real-case examples, but the authors feel it is beyond the scope of this current paper to include the results here.

L.203: "the addition of soil tiles (and therefore more soil columns) has allowed each surface tile to have different rooting profiles and rates of water extraction." How does this harmonise with L.71 "The root density is assumed to follow an exponential distribution with depth, with the depth scale varying between the different PFTs.", which I assume is independent of the soil tiling method ?

Under the default scheme, all the surface tiles (PFTs) access soil moisture through a rooting profile that is distributed through a single soil profile over four layers. In each layer, the soil moisture is shared between all the surface tiles accessing it via the collective root system. In contrast, under the soil tiling scheme, the surface tiles have the potential to access soil moisture through roots which are distributed across multiple soil profiles. For SurfGB, SurfDom and HResTex each surface tile will have its own soil profile each with a different rooting profile and extraction rates associated with it. For HResTexAgg, each surface tile has access to more than one soil profile and therefore soil moisture can be extracted from more than one rooting profile as well. We have modified the manuscript to make these points clearer.

In section 3, the text should be pruned much more stringently to avoid repetitions (see some examples below). Section 4 seems to repeat a lot of section 3, but in a much more readable form. I wonder if section 3 can be shortened significantly, e.g. removing the explanations that are repeated in section 4 (keeping section 4 as is).

Thank you for the helpful suggestion. We have shortened section 3 to make it more concise and readable. We have also removed explanations that are repeated in section 4.

**Technical corrections**

L.41: "Due to the non-linear nature of soil processes, the dominant soil type is used for each grid box and soil parameters associated with this soil type are then used." What would the alternative be when only using one soil tile? A soil type with some sort of weighting of the different soil parameters? Perhaps this is the "aggregated" soil properties used in the CLS and ISBA models, but it reads a bit obscure in the text before this is mentioned a few lines later.

The dominant soil type is used rather than an average soil type in each grid box. We have amended the manuscript to clarify this.

L.69, 72, 191, Fig.4: $\beta$ is called "soil moisture availability factor" on L.69 and in Fig.4 and "soil moisture stress factor" on L.72 and L.191

For consistency $\beta$ will be called the "soil moisture availability factor" throughout the manuscript. We have changed all incidences of "soil moisture stress factor" to be referred to as "soil moisture availability factor" instead.

L.70, 72: The definition of $\beta$ is split into two sentences, surrounding a description of root density. Can the first sentence be merged with the second ?

We have merged the two sentences such that the paragraph now reads as follows (lines 72 - 80): "The flux of water extracted by the vegetation from the soil for transpiration is determined by the root density and the soil moisture availability factor ($\beta$). The

root density is assumed to follow an exponential distribution with depth, with the depth scale varying between the different PFTs. $\beta$ is a dimensionless moisture stress factor...."

L.190: "are gradually become"
Changed to "are gradually becoming".

L.191-197 (and further on). Description of the line colours and styles in the text seems a bit redundant
We have removed the descriptions of line colours and styles from the text. The legend should provide a clear enough explanation on its own.

L.204: $\beta$ written as "beta"
The '\' was missing in the latex maths mode for beta. This has now been added and written as $\beta$.

L.210-212, L.259-260: Repetition of the same information.
I have deleted the repetition of this information from lines 259-260 as I realise it is now no longer needed.

L.222-227: Seems to be a lot of redundant information in these sentences. Compress ?
I have compressed these sentence, so most of the redundant information has been removed.

L.235-237, L.248-249. Repeating more or less the same thing.
I have removed explanations that are repeated and made the paragraph more readable and concise.

L.239-245, L.249-251: Repeating the same thing, but less detailed.
I have removed explanations that are repeated and made the paragraph more readable and concise.

**Community Comments 1**

Interesting work! I have some minor comments for the authors' considerations:
1. Is the testbed grid box from real word or artificial assumptions? Please give more details.
The testbed grid box is artificial and devised to enable us to represent all five different surface types and a full range of soil textures. For further details please see my response to reviewer 1 on a similar question.

2. Whether was the heterogeneity of soil organic matter considered?
We haven't considered soil organic matter here as it is beyond the scope of this work. However, we acknowledge that organic soils should be considered in future studies.

3. How does JULES calculate soil albedo for different soil types?

For the purpose of this study the soil albedo has been kept constant for different soil types. We don't expect the results here to be sensitive to heterogeneity in the soil albedo.

4. Apart from the LSMs listed in Table 1, E3SM land model (ELM) can represent the soil heterogeneity at different topographic units under a novel topography-based sub-grid structure (Hao et al., 2022). Hao, D., Bisht, G., Huang, M., Ma, P.-L., Tesfa, T., Lee, W.-L., et al. (2022). Impacts of sub-grid topographic representations on surface energy balance and boundary conditions in the E3SM land model: A case study in Sierra Nevada. Journal of Advances in Modeling Earth Systems, 14, e2021MS002862. https://doi.org/10.1029/2021MS002862

Thank you for the information, we have added ELM to table 1 and refer to it in the manuscript.

5. I am curious why the lines for SurfDom and SurfGB overlap with the HResTex run in all variables but HResTexAgg has some differences from the HResTex. Please explain it.

We have added the following text to the manuscript in response to a similar question from reviewer 1 above. Hopefully this answers your question: "The SurfGB and Surf-Dom experiments (not shown) have a similar annual cycle to HResTex despite the fact that they use far less soil information and tile the soil according to the surface type (as opposed to using high resolution soils). The reason for this similarity is that they all use a one-to-one mapping between the surface and soil tiles. This results in similar rates of drying, onset of soil moisture stress and resultant latent heat fluxes.". In contrast, for HResTexAgg, each surface tile has access to more than one soil profile and therefore soil moisture can be extracted from more than one rooting profile resulting in different rates of drying.

---

## Referee Report (RR1)

Referee report

The authors provide very good replies to all questions. In particular, the rationale for using a synthetic example case study is better explained in the updated manuscript. One might have wished more of the replies had gone into the text (especially JULES (lack of) demography), but I trust that most readers are better aquainted with the model than I am. I think the updated manuscript is well suited for publication.

---

## Author Response (AR2)

**Reply to the editors' comments: gmd-2022-139**

Heather S. Rumbold et al.

January 26, 2023

We would like to thank the editor for their time to read and comment on the revised submission of this manuscript. We have addressed the final technical corrections and hope this improves the manuscript satisfactorily. The following has been modified since the last revision:

- Results is now the main heading for Sect. 3. The subsequent text has been split up into 4 individual sections.

- The heading numbers in section 2 are now consistent and read as sections 2.1 and 2.2.

- Section 4 now contains just the conclusions and section 5 has been added to describe the applications of use for the scheme and future work.